

# Climatology of ionosphere over Nepal based on GPS TEC data from 2008 to 2018

Drabindra Pandit[1, 6], Basudev. Ghimire[1,6], Christine Amory-Mazaudier[2, 3], Rolland Fleury[4], Narayan Prasad Chapagain[5], Binod Adhikari[6]

1. Central Department of Physics, IOST, Tribhuvan University, Kathmandu, Nepal
2. Sorbonne Universités UPMC Paris 06, LPP, Polytechnique, France
3. T/ICT4D Abdus Salam ICTP, Italy
4. Lab-STICC, UMR 6285, Institut Mines-Telecom Atlantique, site de Brest, France
5. Amrit Campus, Tribhuvan University, Thamel, Kathmandu, Nepal
6. Department of Physics, St. Xavier's College, Maitighar, Kathmandu, Nepal

## Abstract

In this study, we analyze the climatology of ionosphere over Nepal based on GPS derived VTEC observed from four stations: KKN4 (27.80° N, 85.27° E), GRHI (27.95° N, 82.49° E), JMSM (28.80° N, 83.74° E), DLPA (28.98° N,82.81° E) during years 2008 to 2018. The study illustrates the diurnal, monthly, annual, seasonal and solar cycle variations of VTEC during all time of solar cycle 24. The results clearly revel the presence of equinoctial asymmetry in TEC which is more pronounced in maximum phases of solar cycle in year 2014 at KKN4 station followed by descending, ascending and minimum phases. Diurnal variation of VTEC showed short-lived day minimum which occurs between 5:00 to 6:00 LT at all the stations considered with diurnal peak between around 12:00 to 15:00 LT. The maximum value of TEC is observed during spring equinox than autumn equinox with a few anomalies. Similarly, winter anomalies are noticed during increasing and maximum phases of the solar cycle2011 and 2014 from almost all stations taken in the study.

## 1.    Introduction

Total electron content (TEC) is a crucial parameter of ionosphere comprising high concentration of electrons and ions, formed under the ionization of extreme ultraviolet (EUV) radiation and solar X-rays. The lower atmospheric disturbance also contributes to ionospheric variability (Anderson and Fuller-Rowell, 1999; Prikryal et al., 2010). A numerous periodic and aperiodic variability identified in the ionosphere makes impact on the applications involving radio link



between satellites and ground which play vital role in the communication, navigation and
surveillance with important consequences for the reliability and accuracy of the service (Guo et
al., 2015). Global positioning system (GPS) is widely used in recent appliances which encounter
largest errors in the path due to disturbed ionospheric free electrons which emphasizes to study
GPS-TEC variability. The application of GPS technology allows scientists insight into the shape
and behavior of ionosphere. List of factors affecting TEC are ionospheric electron density, ion-
electron temperature, composition, dynamic vary with altitude, latitude, longitude, local time,
seasons, solar as well as magnetic activity. Equatorial ionosphere is being highly vulnerable
possess major threats to the communication signals. The ionosphere at mid latitude is less
variable hence the most of the observations and measurements are taken from this region where
as high latitude ionosphere is sensitive to outer space connected by geomagnetic field lines
(Akala et al., 2013; Parwani et al., 2019). Study of VTEC at low-mid ionosphere showed solar
activity dependence (Shimeis et al., 2014). TEC has been studied by large number of researchers,
Rama Rao et al. (1980) studied the diurnal variation in TEC at Waltair, India and found short
lived pre-dawn minimum, a steep early morning rise followed by broad mid afternoon maximum
and a steep post sunset fall. The relation between TEC and SSN, $F_{10.7}$ and EUV was studied by
Dabas et al. (1993) and pointed out that TEC has nonlinear relation with SSN and linear relation
with $F_{10.7}$ and EUV. Ouattara and Amory- Mazaudier (2012) showed solar activity occurring
different phases of solar cycle in diurnal variability. Analogous study was carried around the
globe using various methods on TEC such as diurnal, monthly, seasonal and solar cycle and solar
activity dependency e.g. in South Asia (Chauhan et al., 2011; Walker et al., 1994); in South
America (Sahai et al., 2007; Natali and Meza, 2011; Akala et al., 2013; de Abreu et al., 2014 );
over North America (Huo et al., 2009; Perevalova et al., 2010); in Africa (Shimeis et al, 2014;
D'ujanga et al., 2012; Ouattara and Fleury, 2011; Zoundi et al., 2012); over Brazil (Venkatesh et
al., 2014a, 2014b, 2015) ; over Japan (Zakharenkova et al., 2012; Mansoori et al., 2016); over
China (Guo et al., 2015; Zhao et al., 2007 ; Liu et al., 2013).
TEC studied at Jet Propulsion laboratory for the year (1998-2008) found stronger annual TEC
variation in southern hemisphere and variation in phase and amplitude is more in conjugate
hemisphere (Liu et al., 2009). Galav et al. (2010) found semiannual periodicity in daytime TEC,
and the spring equinox shows highest TEC and winter solstices the lowest in India. The winter





anomaly, semiannual anomaly and annual anomaly are described in paper by Liu and Chen
(2009); Rishbeth and Garriott (1998). Global scale TEC research found that the effect on TEC
was stronger on day than at night and also at low latitude than in high latitudes. The effect on
TEC is seen more on the either side of dip equator than at dip equator (Liu et al., 2009). Parwani
et al. (2019) studied latitudinal variation of ionospheric TEC at northern hemispheric region and
found that the diurnal TEC has higher value in low than in mid and high latitude and in seasonal
variation maximum in Spring and autumn than in summer and winter.
Many studies on TEC have conducted in Asia, however no result for the climatology of TEC
over Nepal for a long-time series, about one solar cycle has been reported up to now. In this
paper, we present for the first-time characteristics of ionosphere in Nepal such as the diurnal,
annual, seasonal and solar cycle dependence of TEC on the local ionospheric conditions using
GPS TEC data obtained from four GPS stations: KKN4, GRHI, JMSM and DLPA. Our study
includes GPS TEC data from 2008 to 2018 of the solar cycle 24, including all four phases of this
solar cycle, the minimum phase of year 2008-2009; ascending phase of year 2010-2013,
maximum phase of year 2014 and descending phase of year 2015-2018. The second section of
this paper includes the dataset and methodology, the third for the results and discussion. The
concluding remark is discussed in the last section.
## 2.  Data sets and data analysis
Total electron content (TEC) is total number of electrons integrated along the path from receiver
to each GPS satellites which orbits the Earth at altitude of 20,200 km. It measures in TECU,
$1\text{TECU} = 10^{16}$ electron/m$^2$. The TEC is obtained as (Hofmann-Wellenhof et al., 1992)
$$\text{TEC} = \int_R^S N_e(h)\,dh \qquad (1)$$
Where, $N_e$ is electron density, R is the receiver altitude and S the satellite altitude. The dual
frequency GPS receiver in two L-band of frequency: $f_1 = 1575.42$ MHz and $f_2 = 1227.60$ MHz
provide the carrier phase and pseudo-range measurements. The TEC is calculated from these L1
and L2 pseudo-range and carrier phase (Hofmann-Wellenhof et al., 1992). Using pseudo-range
and phase data, TEC is calculated as





$$\text{TEC} = \frac{1}{40.3} \left( \frac{f_1{}^2 f_2{}^2}{f_2{}^2 - f_1{}^2} \right) (P_1 - P_2) \qquad (2)$$

Where, $P_1$ and $P_2$ are the pseudo-ranges for frequencies $f_1$ and $f_2$, respectively.
The TEC obtained by this method is called slanted TEC (STEC) which is a measure of the total
electron content of ionosphere along the ray path from the satellite to receiver has to be
converted to vertical TEC (VTEC) using equation (Titheridge, 1972).

$$\text{VTEC} = (\text{STEC} - B_s - B_u) \left( \sqrt{1 - \left( \frac{(R_e \times cos\in)^2}{(R_e + h)^2} \right)} \right) \qquad (3)$$

$B_s$ and $B_u$ are the biases of instruments of satellites and receivers respectively, $\in$ is the elevation
angle of satellite and $R_e = 6371$ km is the mean radius of the Earth.
For this study data was carried out with GPS data taken from four GPS stations: DLPA, JMSM,
KKN4 and GRHI from Nepal. The details of the stations including their geographical and
geomagnetic coordinates are shown in table 1 and Universal time is considered as all-time
references. The GPS data of the four stations were downloaded from www.unavco.org which is
freely available to all users. This data is available in RINEX (Receiver Independent Exchange
format) v2.1 which is a standard ASCII format. The temporal resolution of this data is 15 min.
The raw data is then processed using software developed by Rolland Fleury (Rolland Fleury,
July 19, 2018) from Lab-STICC, UMR 6285, Institut Mines-Telecom Atlantique, site de Brest,
France which runs on a window operating system to get required TEC.
The data for solar indices sunspot number (SSN) and solar flux index (F10.7) to study long term
solar activity are taken from Royal Observatory of Belgium, Brussels, through website:
sidc.oma.be/silso/home and OMNI website http://omniweb.gsfc.nasa.gov/. SSN is most
consistent solar indices effectively describes solar activities and are valuable mode in forecasting
space weather phenomena. The solar flux index provides the information about the total emission
produced by the Sun at the wavelength of F10.7 cm at the Earth.
In this study, we use GPS derived TEC from RINEX file using this method to obtain TEC
calibrated at 15 minutes for all measures. Between 30s VTEC sequences, the elevation may vary.
This leads to variation in the VTEC depending on the constellation and not just the variation of





the content over that period. We have chosen to do the regression over a period 15 minutes with
the VTEC obtained displayed in the middle of this period. This makes it possible to have 4 points
over 1 hour and therefore to have an evolution of the VTEC 4 times more precise than that of
GIM maps which are currently in steps 1 or 2 hours depending on the organization. So, it is
better possibility to see and characterize finer local structures in RINEX derived TEC than in
GIM.
This study analyzes variations of VTEC during different phases of solar cycle 24 along with
annual, seasonal and diurnal variation. For this local season classified as winter (November,
December, January and February), spring (March and April), summer (May, June, July and
August) and autumn (September and October). The classifications of selected years as per solar
cycle phases are presented in table 2.

## 3. Results

In this section, we present the signatures of diurnal, monthly, seasonal, solar cycle and
geomagnetic variation on GPS VTEC over Nepal which is calculated using Rolland software (as
described in sec. 2). Figure 1 represents the position of GPS TEC stations in Nepal used for this
study and figure 2 represents the variation of sunspot number and solar flux for the year 2008 to

18    2018.

### 3.1 Diurnal variation

Figure 3a exemplify diurnal variation of VTEC in LT time observed during February 2, 2009,
2012, 2014, 2016 and 2017during the minimum, increasing, maximum and decreasing phases of
solar cycle 24 at KKN4 station in Nepal. The plot shows before sunrise ~5:00 LT, VTEC
becomes minimum and reaches a maximum around 13:00-14: 00 LT and later decreases in the
evening and night. The diurnal peak is noticed between 12:00 to 15:00 LT though the peak
values change every month. The VTEC plots reveal a growth from dawn to a highest value about
5 to 47 TECU after the hours of daylight, it decreases to the lowest value prior to dusk with time
difference of ±1 to 2 hours. A flat curve with minor peaks is identified during minimum and
descending phases whereas dome shape with multiple peaks and trough at varying position is
observed during ascending and maximum phases. In overall, the VTEC shows a normal trend of
diurnal behavior with the lowest value in dawn and dusk and the highest value during the



midday. The maximum VTEC in diurnal curve noticed during maximum phases of the solar
cycle 2014 then in 2012 the ascending phases and minimum in 2016, 2009 and 2017 during
descending and minimum phases. Diurnal variation of VTEC was studied by plotting similar
curves for all the days from year 2008 to 2018 for all chosen four stations. In general, the diurnal
VTEC behavior exhibits the solar cycle dependency. The diurnal variability of VTEC for all the
day is not presented due to constraint of space. In our study the mean diurnal curves for KKN4
station of year 2008, 2009, 2010 and 2017 exhibits wave like profile whereas the mean diurnal
curves of year 2011, 2012, 2013, 2014, 2015 and 2016 shows parabolic nature which is shown in
fig 3b. The similar diurnal profile was noticed for all the stations considered.
### 3.2 Monthly variation in TEC
Figure 4 shows the monthly variability of VTEC for the maximum phase of solar cycle year
2014 at KKN4 station. The plot is obtained using average of daily data. The plot shows
maximum in equinoctials months (March, April) and minimum in solstices (January). The rise or
fall of TEC in each curve follows the diurnal pattern, prominent peak in the midday with
different peak amplitude. The lowest VTEC peak observed during January and highest in March.
Late afternoon peak noticed in March, June and September whereas the peak centered ~ 2:00 LT
for rest of the months. A significant plat peak noticed in December whereas the steep rise in
VTEC is noticed in March, April and October. Monthly variation of VTEC was studied by
plotting similar curves for all the month from year 2008 to 2018 for all chosen four stations. The
plot shows clear wave's activity in mean diurnal curve for year 2008, 2009 and 2010 and from
the years 2011 to 2016 the stiff rise in VTEC was noticed and in 2017 the wave activity starts
again (plots are not included in this paper).
### 3.4 Seasonal variation in TEC
Figures 5 show two dimensional diurnal plot of VTEC at JMSM station for all the four phases (I
minimum-2009, II ascending-2011, III maximum-2014 and IV descending-2015) of the solar
cycle 24 which explains how diurnal VTEC varies hourly during four phases. From fig. 5a, 5b,
5c and 5d it is observed that equinoctial asymmetry is not noticed in 2009, in 2011 autumn is
more intense than spring and in 2014 and 2015 spring VTEC is greater than autumn. In the fig.
6a, 6b, 6c, 6d and 6e each panel separately represents the VTEC variation during autumn, spring,



summer and winter season for the year 2008, 2009, 2011, 2014 and 2015 at KKN4, GRHI,
JMSM and DLPA respectively. The plots show the maximum values of VTEC is ~95 tecu in
spring 2014 the maximum year of sunspot cycle and minimum value 10 tecu in 2009 winter the
minimum year of sunspot cycle. In the increasing and decreasing phases of solar cycle, the
VTEC gradually increases and decreases depending to the amount of UV that arrive the Earth. In
general, the plots show that VTEC is maximum during spring followed by autumn, summer and
winter, except few cases. Similarly, previous study of GPS TEC for the year 2014 over Nepal
also reported the highest value of VTEC on March and lowest on December with distinct the
seasonal variations having higher values in spring and lower in winter season (Ghimire et al.,
2020). During the sunspot minimum years 2008 and 2009 and 2010, there are no semi-annual
variations in the VTEC and it also seems the summer VTEC is as strong as the Autumn VTEC.
For the years 2011 to 2016 the semi-annual variations are noticed. During the year 2017, we
observed the same pattern as for the year 2008, 2009 and 2010, the summer VTEC is as strong as
the Autumn VTEC. At the station KKN4, the VTEC in autumn is very weak in year 2015 and it
is smaller than the VTEC in summer. In year 2011 winter anomaly is noticed at KKN4 whereas
GRHI and JMSM winter anomaly is observed in year 2011 and 2014. Winter anomaly is not
observed at DLPA. In year 2008 spring VTEC identified more than autumn for GRHI, JMSM
and DLPA but less than autumn is observed KKN4. In year 2009 only at JMSM spring noticed
greater than autumn. The autumn VTEC is greater than spring for all station in 2011 except at
JMSM it equal to spring. Large asymmetry is noticed between spring and autumn in year 2014.
In year 2015 the summer peak is higher than autumn. In fig. 7 top left panel represents the
variation of VTEC during spring, down left during autumn, top right during summer and bottom
right during winter from year 2008 to 2017 at KKN4. In spring the difference in VTEC between
high and low solar activity is 65 tecu, in autumn 53 tecu, in summer 45 tecu and in winter 40
tecu respectively. In fig. 8 the top panel represents VTEC variability during minimum and
increasing phases whereas the bottom panel represents the maximum and decreasing phases of
solar cycle 24 using GPS station at GRHI. The plot shows that equinoctial asymmetry is not
observable during minimum solar cycle 2008 and 2009, it is clearly distinguishable during other
phases of solar cycle. In general, the sunrise time in summer and winter is 5:15 LT and 6:45 LT
which are differ by 1.5 hours. During summer 2014, the maximum and minimum TEC observed
is 21 and 12 tecu whereas in winter the maximum and minimum tec noticed is 25 and 15 tecu



respectively (figure 4). It also seems that during sunrise time in summer the VTEC is linear but
during the winter it is steep.
### 3.3 Solar cycle variation of TEC
Figure 9 shows the annual mean values of VTEC, solar flux index and sunspot number during
the solar cycle from year 2008 to 2018. The black, blue, green and red color line represents
VTEC variation on station KKN4, GRHI, JMSM and DLPA whereas pink and light green color
line represents variation in SSN and solar flux index, respectively. The plot shows VTEC
gradually begins to increase in 2009 and reaches a maximum in 2014.Then it begins to decrease
till 2018 which agrees with the sunspot number and solar flux variation in the same plot. The
figure shows that the maximum value of peak of ionization in 2014 about 37 tecu in maximum
phase of solar cycle and the minimum value in 2008 about 11 tecu in the minimum phase of solar
cycle. The observed VTEC variation corresponds to the amount of UV reach to the Earth.
Similarly, the solar flux increases from 2011 onward; the measured VTEC also exhibits highest
magnitude for the year 2014. The maximum VTEC value shows a decreasing trend since year
2015 to 2018 at all the stations used for this study. It is observed from the graph that average
annual VTEC shows better synchronization with SSN and solar flux index.
### 3.4 Effects of geomagnetic activity on TEC
Our discussion has so far been confined to the diurnal, monthly, seasonal and solar cycle
variation during solar cycle 24 but the presence of geomagnetic storms indicates solar wind-
magnetospheric interaction which originates global disturbances in the Earth's magnetic field as
well as in ionospheric TEC. The change of solar and geomagnetic activity during the solar cycle
produces variation in all ionospheric parameters (Bremer, 2004). During this study, we find
VTEC variation by severe ($Dst \leq -200\ nT$), intense (-100 nT$\leq Dst \leq -200\ nT$), moderate (-
50 nT$\leq Dst \leq -100\ nT$) and week (-30 nT$\leq Dst \leq -50\ nT$) storms (Gonzalez et al., 1994). We
leave their detail study for the future work.
## 4. Discussion
### 4.1 General background





Since the first survey of the ionosphere by Breit and Tuve in 1926, many surveys have been done
with ionosonde to characterize the electron density of the ionosphere mainly the maximum of
electronic density of the F2 layer : NmF2 (Rishbeth and Garriott, 1969). Since two decades,
GNSS receivers have provided access to the TEC of the ionosphere. NmF2 and VTEC provide
information on variations in ionospheric ionization.
Solar EUV, UV and X rays are the main source of ionization at low and mid latitudes on the
dayside of the earth. Earth's rotation introduces a diurnal variation in ionization. The ionizing
rays (EUV, UV and X) are emitted on the sun depend upon the sunspots. The sunspots have a
cycle of about 11 years. Ionization of the ionosphere follows this 11-year cycle.
Morphological characteristics of the ionosphere based on the NmF2 are known since several
decades:
- annual and semi-annual variations, with stronger electron densities at the equinoxes (Rishbeth
et al., 2000)
- winter anomaly in F2 region electron densities with greater daytime electron densities at the F2
peak NmF2 in winter than in summer ( Rishbeth and Garriott, 1969)
- ionospheric equinoctial asymmetry defined as different ionospheric behaviour in the two
equinoxes
- night maximum of ionization related to the post sunset $\vec{E} \times \vec{B}$ drift (Sastri,1983).
In this section, we will discuss our results presented in section 3 at the light of the different
physical processes creating the variations in ionization.

## 4.2 Sun radiations and solar cycle

The patterns of the solar cycles play a major role in the solar variability: solar radiation and
sunspot number and consequently influence the ionosphere. The solar cycle 24 is the smallest
solar cycle since the spatial era (1957), in which peak is noticed in 2014, some few major solar
flares were erupted from the Sun in February and October 2014 (Kane, 2002) so the maximum
VTEC is noticed in February, October shown in fig. 8. Again from fig. 8, higher value of sunspot
and solar flux was reported in February 2011 corresponding to X-class solar flare at which
higher value of VTEC noted in station considered.
Sharma et al. (2012) studied VTEC variation at Delhi lies near equatorial crest region during
year 2007 to 2009 low solar activities and found TEC has a short-lived day minimum between

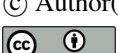



5:00-6:00 LT and gradual increase and reaches its peak value between 12:00 to 14:00 LT. The
day minimum was found flat during most of the nighttime hours (22:00 to 06:00 LT).Their
results show magnitude of daily maximum TEC decreases since 2007 to 2009 due to decrease in
solar flux. They also found TEC seasonal behavior depends on the solar cycle and the largest
daily TEC is observed during equinoctial month at Delhi. In 2020, Ghimire et al, studied diurnal
variation of TEC at JMJG (Lamjung, Nepal) station for the year 2015 found the minimum in pre-
dawn, a steady increase in the early morning followed by afternoon maximum then gradual
decrease after sunset the similar pattern is also observed our study.
In African sector, Tariku (2015) observed from 2008 to 2009 and high 2012 to 2013 values of
VTEC during the low and high solar activity phases. According to their finding, the diurnal
VTEC values attained maximum in the time interval of 13:00 to 16:00 LT and the least values
are mostly at around 06:00 LT.
The similar result is noticed in all considered Nepalese GPS stations during the low solar active
phases of solar cycle 24 in Nepal. The maximum diurnal variability in VTEC in 2014 is caused
by solar active period confirmed by maximum sunspot number (SSN) and solar flux index
(shown in fig. 2), VTEC greater in 2012 due to second maximum in SSN and solar flux and
minimum VTEC in 2009 and 2017 supported by minimum SSN and solar flux. The diurnal
graphs (fig. 3) show better synchronization of VTEC with SSN and solar flux (fig. 2).
In Nepalese ionosphere the diurnal VTEC maximum occurs approximately between 12:00 to
14:00 LT. Similar to Delhi station in Nepal, the day minimum was found flat during most of the
nighttime hours (22:00 to 06:00 LT).In general, the value of diurnal peak in VTEC is maximum
during the spring equinoxes except in 2011 in which autumn VTEC is maximum. As the solar
flux decreases from 2008 to 2009, the daily maximum VTEC values show a decreasing trend.
The seasonal variation of VTEC shows a semiannual pattern, maximum in equinoxes and
minimum in summer and winter. It is evident from the fig. 6c and 6d the semi-annual anomaly is
noticed during 2011 and 2014.
**4.3 Diurnal shape**
The observed diurnal VTEC pattern reflects the different solar events signature. The noon bite
out profile with asymmetric peaks, parabolic profile and wave profile with morning, evening and
night peaks and few complex structures are noted in diurnal profile. The quiet day activity at



minimum phase, the fluctuating activity during increasing phase, shock activity during maximum
phase and recurrent activity during declining phase was noticed in the study of ionospheric
parameters at Ouagadougou ionosonde station data in West Africa by Ouattara et al. (2009).In
mean diurnal curve of year 2008, 2009, 2010 and 2017 the wave like profile and from 2011 to
2016 the parabolic profile were noticed in Nepalese VTEC (figure 3b).

## 4.4 Semiannual variation, equinoctial asymmetry

The important parameter for semiannual variation of ionospheric ionization is the variation in
atomic/molecular ratio, i.e. concentration of $O/N_2$ ratio. At solstice, there is circulation of
meridional wind of about 25 m/s in middle and low latitudes from summer to winter hemisphere
(Rishbeth et al., 2000). These winds carry nitrogen-rich air produced in summer hemisphere into
lower latitudes by upwelling in higher latitudes, reducing $O/N2$ ratio. At equinox, there is no
prevailing meridional circulation. The ratio $O/N2$ depends specially on the horizontal circulation,
and its seasonal changes accompany the change in global thermospheric circulation between
summer-to-winter pattern around the solstices to a symmetrical pattern at equinoxes. The six
possible reasons of seasonal and semiannual variations in $F_2$ layer discussed by Rishbeth (1998)
are:  a) The compositional changes due to large-scale dynamical effects in the thermosphere b)
Variations in the geomagnetic activities c) Energy of solar wind d) The inputs from lower
atmospheric phenomena such as waves and tides e) change in atmospheric turbulence and f)
Anisotropy of solar and EUV emission in solar latitude (Burkard, 1951).
In 2020, Ansari et al. found the minimum value of TEC in January and that becomes maximum
in April then decreases in June-July and followed by increase in magnitude of second maximum
in September-October and later decrease down till December at CHML and JMSM GRHI of year
2017. Referring figure 8, our result of semiannual variation shows the minimum value of VTEC
is found in January and that becomes maximum in March-April then decreases  in June-July and
followed by increase in magnitude of second maximum in October-November and later
decreases down till December at GRHI of year from 2009 to 2018.
The asymmetry between the two equinoxes is due to geophysical parameters as magnetic indices
related to geomagnetic activity (Triskova, 1989) and the IMF Bz the interplanetary component of
magnetic field (Russell and Mc Pherron, 1973). The equinoctial asymmetry observed in VTEC is

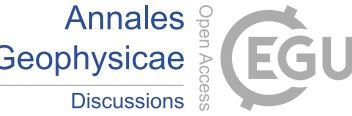

explained by i) the axial hypothesis ii) the Russell Mc Pherron (RM) effect and iii) the
equinoctial hypothesis (Chaman Lal, 1996; Shimeis et al., 2014).
Ouattara and Amory-Mazaudier (2012) made a statistical model of the F2 layer, at equatorial
latitudes, based on data obtained during 3 sunspot cycles. This model shows the influence of the
different type of geomagnetic activity defined by Legrand and Simon (1989) and the asymmetry
of equinoxes due to the magnetic activity. The asymmetry between the two equinoctial peaks is
also due to asymmetry of the thermospheric parameters that influence ionosphere as neutral wind
and change in composition (Balan et al., 1998)
In Nepalese ionosphere features equinoctial asymmetry is distinctly noticed in 2D plots of year
2009, 2011, 2014 and 2015 in fig 5b, 5c and 5d, respectively. In year 2009, equinoctial
asymmetry is not noticed during low solar activities. But in year 2011, the autumn is intense than
spring which is the features of equatorial ionization anomaly (EIA) crest latitude and in year
2014 the difference between equinoctial asymmetry is less (spring > autumn) which is again the
characteristics of the EIA trough station. And in 2015, the asymmetry very high (spring >
autumn) which is the general feature of TEC at all latitude. Our study can conclude that Nepalese
ionosphere sometimes show features of EIA crest latitude and sometimes EIA trough station.

## 4.5 Winter anomaly

The winter or seasonal anomaly introduced due to temperature changes (Appleton, 1935), inter
hemispheric transport of ionization (Rothwell, 1963), the significant changes in the Sun-Earth
distance (Yonezawa, 1959), seasonal variation of O/N2 concentration (Rishbeth and Setty, 1961;
Wright, 1963; Rishbeth et al., 2000; Zhang et al., 2005) and the upward movement of energy
flux (Maeda et al., 1986). The winter anomaly is related to solar activity. Tyagi and Das Gupta
(1990) and Bagiya et al (2006) have reported absence of winter anomaly in low solar activities at
low latitudes. The change in composition of the constituents being identified as cause of the
winter anomaly is coined by Rishbeth and Setty (1961). The least VTEC in June solstice (in
Northern hemisphere) during the low and high solar activity phase may be due to the asymmetry
heating and which result in transport of neutral constituents from summer to winter hemisphere
reducing the rate of recombination. The reduction in recombination rate in winter causes the rise
of VTEC in winter than in summer. Gupta and Singh (2001) studied TEC over Delhi and
concluded that winter anomaly in TEC appears only during higher solar activity. This winter





anomaly is due to the closer distance of the Earth from the Sun and the direction of the wind
from the summer season to the winter (Shimeis et al, 2014). Krankowsky et al. (1968) and Cox
and Evans (1970) separately pointed out that the ratio of $O/N_2$ become twice in winter than in
summer as a result of higher electron loss rate in summer than in winter. Torr and Torr (1973)
observed the winter anomaly in $foF_2$ under different solar activity at the mid latitude of northern
hemisphere and similar result was observed in southern hemisphere during high solar activity.
Furthermore, they noticed lower solar activities results lower winter anomaly. In general, June
solstice anomaly is higher than the December solstice but in earlier, study done at Agra GPS
station noticed some abnormalities in the solstice behavior demonstrating higher VTEC in the
summer than autumn and winter anomaly with higher VTEC than in summer (Bagiya et al.,

11 2011).

In present study the winter anomaly is noticed in 2011 and 2014.At KKN4 station winter
anomaly is noticed in the year 2014; at GRHI 2014 and 2016; at JMSM 2014 and 2016. Winter
anomaly is not noticed at DLPA (figure 6c and 6d).

## 4.6 Nighttime maximum

The upward $\vec{E} \times \vec{B}$ drift velocity plays an important role in producing the nighttime post sunset
enhancement. The average plasma flux required for the enhancement in equatorial latitude found
$(2.2 \pm 0.9)$ x$10^{12}$ m$^{-2}$s$^{-1}$ by Jain (1987) in India. In 2015 Tariku, studied pattern of GPS-TEC over
African sector during 2008 to 2009 and 2012 to 2013 and found small enhancements in the
VTEC in the nighttime ~ between 21:00 to 23:00 LT  especially for equinoctial months and then
drops again mostly after 23:00 LT. The enhancement was mostly found in equinoctial months
during high solar activities and during low solar activities phase in solstice the pre-reversal
enhancement was much smaller. A diurnal plot (fig.3) of Nepalese ionosphere shows similar
result of pre-reversal enhancement during high solar activities 2012 and 2014 but not during low
solar activities of 2009 and2017.

## 5. Conclusion

This paper investigates the diurnal, monthly, seasonal and solar cycle variations of VTEC at four
mid low latitude stations: KKN4 (27.80° N, 85.27° E), GRHI (27.95° N, 82.49° E), JMSM
(28.80° N, 83.74° E) and DLPA (28.98° N, 82.81° E) in Nepal.



The following conclusions are found:
- The shape of mean diurnal variation of VTEC depends on the solar cycle phases: no diurnal
peak is observed during minimum and descending phases of the solar cycle whereas a Gaussian
with different peak amplitude is noticed during ascending and maximum phases of the solar
cycle.
- The week ionospheric activities characterized by lower TEC values during minimum and
strong activities by higher value of VTEC during maximum phase.
- Day to day variation in VTEC is significant in all the station. The maximum is noticed at
KKN4 and minimum at DLPA.
- The study may reveal that diurnal TEC maximizes at around 12:00 LT to 15:00 LT, with
minimum in the pre-dawn periods.
- The mean diurnal profile in the year 2008, 2009, 2010 and 2017 exhibit wave like nature
whereas the parabolic nature is observed in the year 2011, 2012, 2013, 2014, 2015 and 2016.
- Equinoctial asymmetry in peak is noticed in spring (March, April) and autumn (September,
October) in which higher is observed during spring.
-The winter anomaly is observed in all the available stations at the maximum of sunspot cycle
2014 and in one other station during the year 2011.
-During the year 2009 of the sunspot minimum the winter anomaly is not observed for all the
stations. And there is no equinoctial asymmetry i.e. very weak (compare to the year of the
maximum) except at JMSM.
-The spring-maximum is smaller than autumn-maximum mainly during years 2011, 2012, 2013
and also during year 2008 for one station, these years are years of minimum or increasing phase
of the sunspot cycle.
-The equinoctial asymmetry is noticed in all the available stations due to difference in the
F10.7cm for the two equinoxes.
-It seems that in Nepal for some years there is no semiannual variation, as we observe sometimes
that summer values are larger than autumn. It is probably a characteristic of Nepal.
**Acknowledgements**
We acknowledge www.unavco.org,http://aiuws.unibe.ch/ionosphere (CODG),



www.ngs.noaa.gov/CORS/Gpscal.shtml,          and          http://www.isgi.unistra.fr/,
http://celestrack.com/GPS/almanar/Yuma/2017/, website: sidc.oma.be/silso/homeand Omni data
sitehttp://omniweb.gsfc.nasa.gov/ for providing RINEX data for TEC, DCB  file, Yuma file and
data for solar wind parameters and geomagnetic indices for our calculations. The author would
like to acknowledge Nepal Academy of Science and Technology (NAST), Nepal for proving
PhD scholarship and ICTP, Italy for giving the opportunity to participate in a workshop on Space
weather effects on GNSS operations at low latitudes.





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





| | |
|---|---|
| 1 | Tables Captions |
| 2 | |
| 3 | |
| 4 | Table 1. The selected GPS stations and their coordinates, the data of which are used in the study |
| 5 | Table 2. Classification of selected years according to the solar cycle phases |
| 6 | |
| 7 | |
| 8 | |
| 9 | |
| 10 | |
| 11 | |
| 12 | |
| 13 | |
| 14 | |
| 15 | |
| 16 | |
| 17 | |
| 18 | |
| 19 | |
| 20 | |
| 21 | |
| 22 | |
| 23 | |
| 24 | |
| 25 | |
| 26 | |
| 27 | |
| 28 | |
| 29 | |
| 30 | |
| 31 | |



1                       List of figures

Fig.1. A map of Nepal showing locations of GPs stations used in our study
Fig. 2 Display the variations of sunspot numbers and solar flux for the year 2008 to 2018.
Fig. 3a. Diurnal variation of vertical TEC in LT at KKN4 GPS station. The black, blue, light
green, red and pink color line represent diurnal variation for the year 2009, 2012, 2014, 2016 and
2017 respectively.
Fig. 3b Diurnal variation of vertical TEC in UT at KKN4 GPS station. The first panel represents
wave like mean diurnal curves of year 2008, 2009, 2010 and 2017 and the second pan lrepresents
parabolic nature of mean diurnal curves of the year 2011, 2012, 2013, 2014, 2015 and 2016. The
plots are arrange following on their diurnal profile.
Fig. 4. Monthlyvariation of vertical TEC in LT for each month of year 2014 at KKN4 station.
Fig. 5 (a), (b), (c) and (d). Two dimensional (2D) variation of vertical TEC according to UT at
JMSM stations for one of the year of minimum (2009), ascending (2011), maximum(2014) and
descending (2015) phases of solar cycle 24.
Fig. 6. Seasonal variability of VTEC during year 2008,2009,2011,2014 and 2015 for KKN4,
GRHI, JMSM and DLPA stations.
Fig. 7. Mean yearly seasonal variation of VTEC for year 2008 to 2017 at KKN4
Fig.8.Maximum VTEC variability at GRHI stations during minimum, increasing, maximum and
decreasing phases of solar cycle 24
Fig. 9. Annual mean VTEC variability at KKN4, GRHI, JMSM and DLPA stations with SSN
and solar flux during year 2008 -2018





1 Table 1

| SN | ID | Locations | Geog. Lat. | Geog. Long. | Geom. Lat. | Geom. Long. | Dip Lat. | Local Time (LT) |
|---|---|---|---|---|---|---|---|---|
| 1 | KKN4 | Kakani, Nepal | 27.80° N | 85.27° E | 18.62°N | 159.41° E | 43.86 | UT+5:45h |
| 2 | GRHI | Ghorahi, Nepal | 27.95° N | 82.49° E | 18.94°N | 156.82° E | 44.25 | UT+5:45h |
| 3 | JMSM | Jomsom, Nepal | 28.80° N | 83.74° E | 19.71°N | 158.06° E | 45.31 | UT+5:45h |
| 4 | DLPA | Dolpa, Nepal | 28.98° N | 82.81° E | 19.94°N | 157.21° E | 46.03 | UT+5:45h |



1    Table 2.

| Interval | Years | Solar cycle phases |
|---|---|---|
| I | 2008, 2009 | The minimum of solar cycle 24 |
| II | 2010, 2011 | The increasing phase of solar cycle 24 |
| II | 2012, 2013, 2014 | The maximum phase of solar cycle 24 |
| IV | 2015, 2016, 2017, 2018 | The decreasing phase of solar cycle 24 |



2    Figure 1

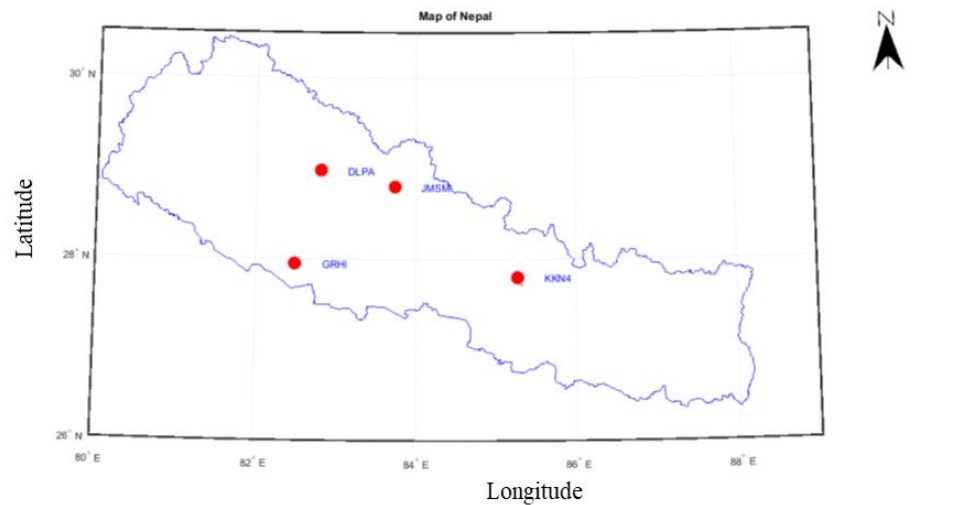





2      Figure 2

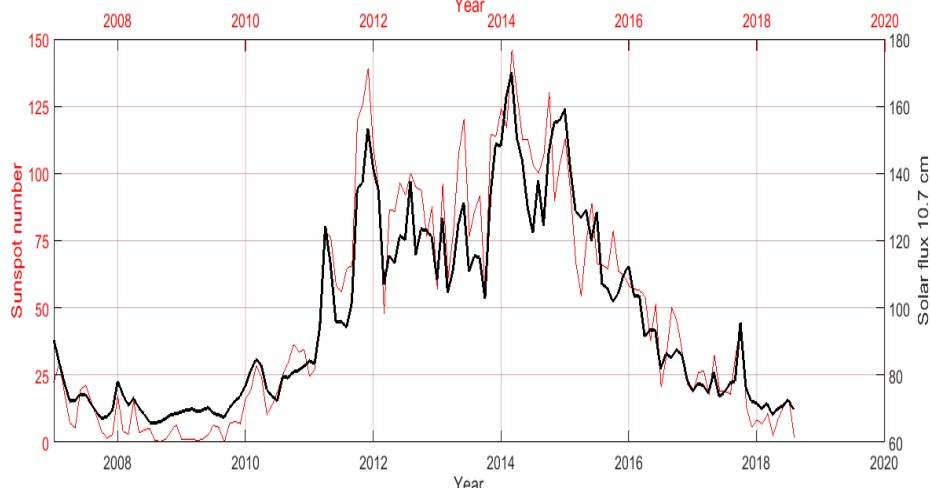



2    Figure 3a

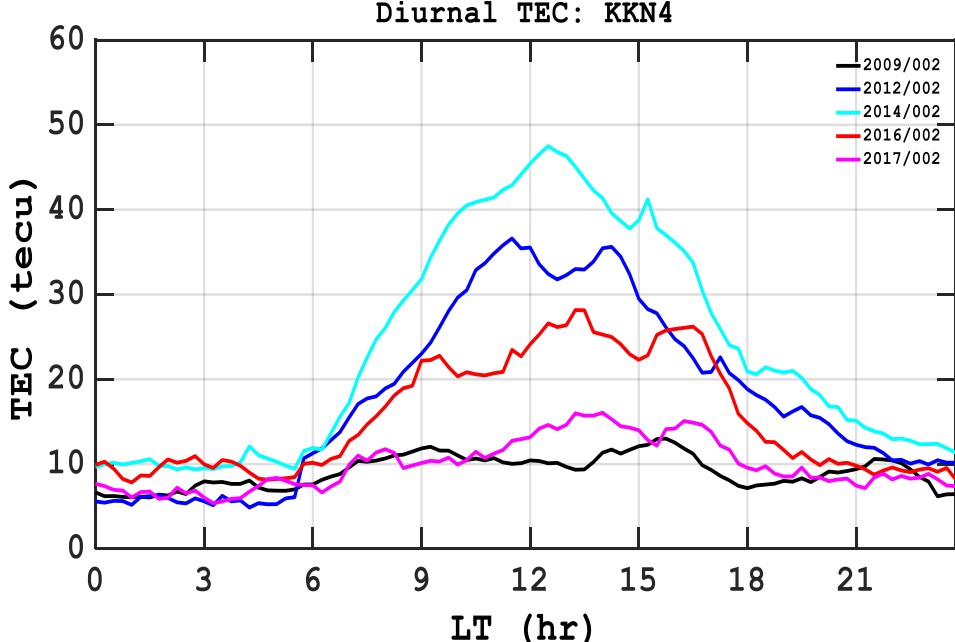





2   Figure 3b

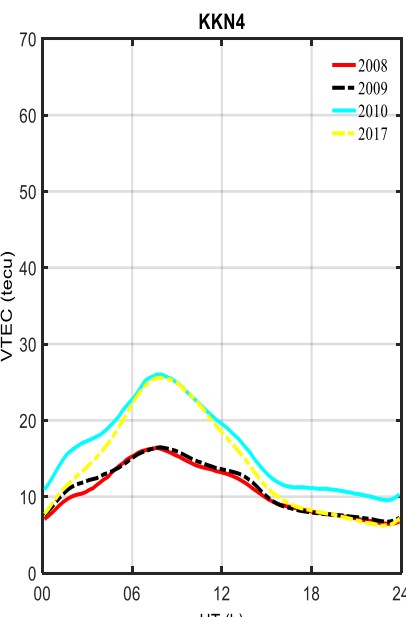 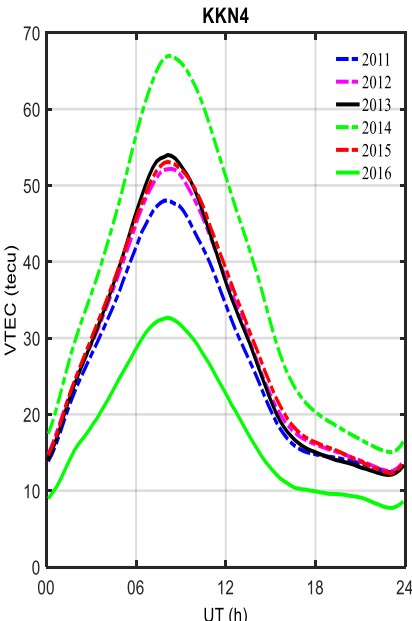

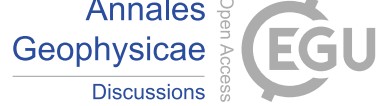

2    Figure 4

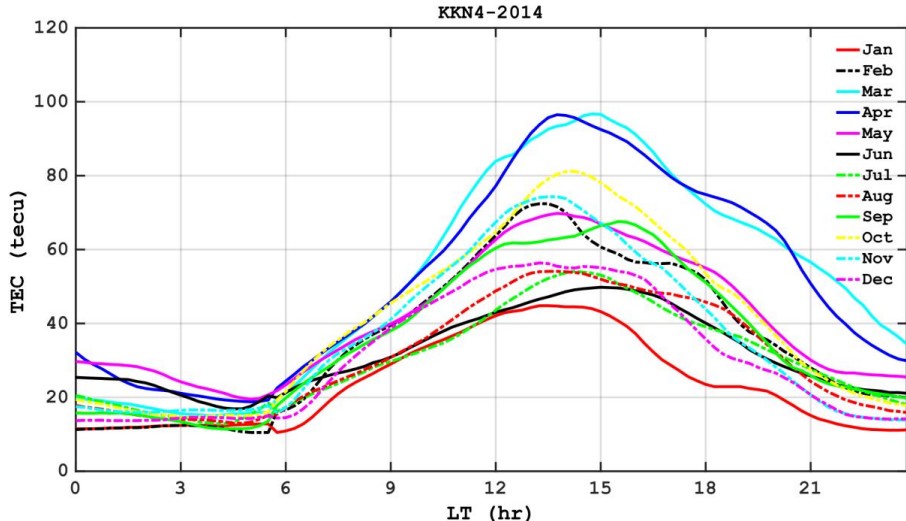



2      Figure 5a

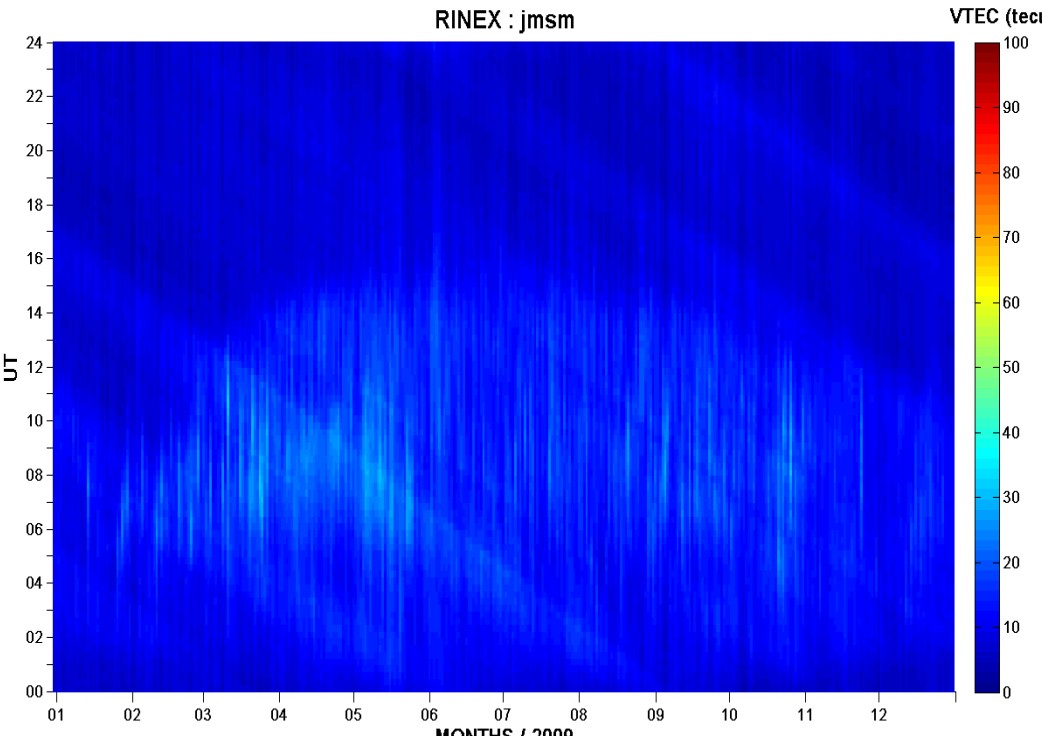



2      Figure 5b

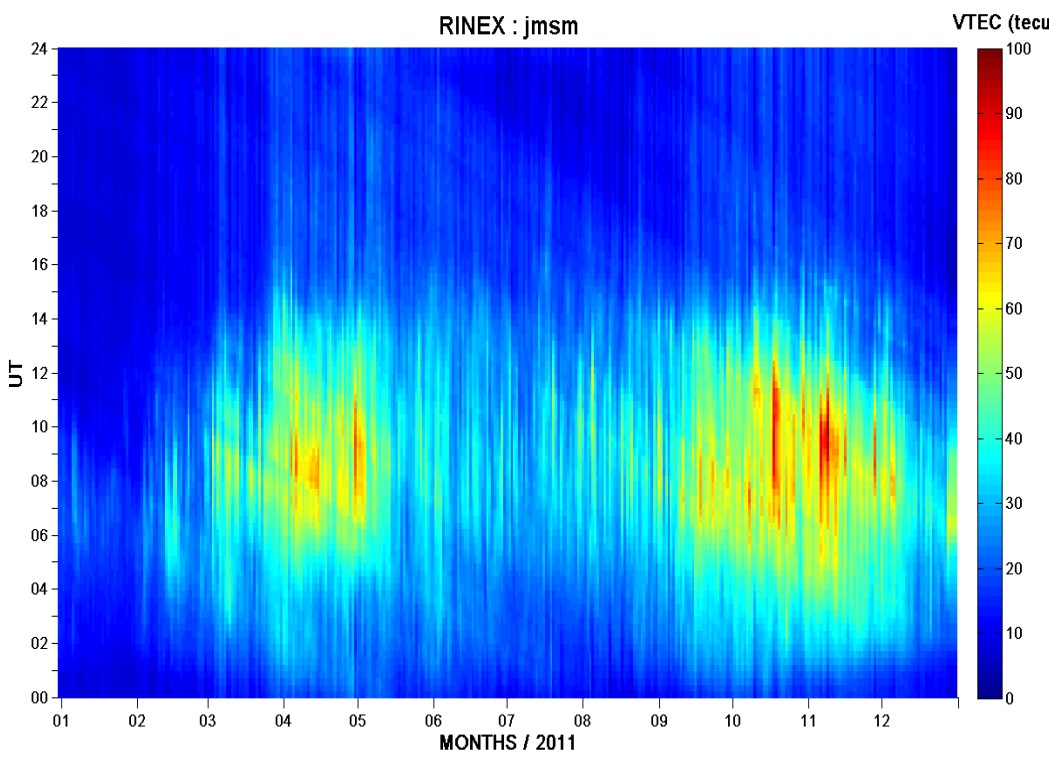





2    Figure 5c



2      Figure 5d

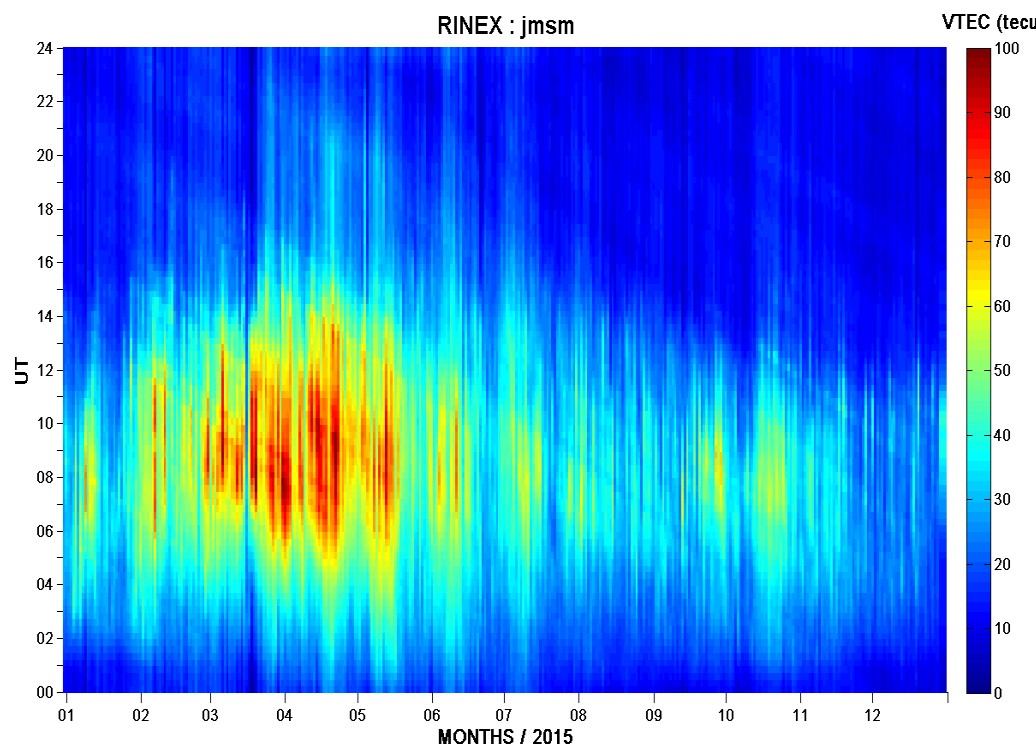





3    Figure 6a

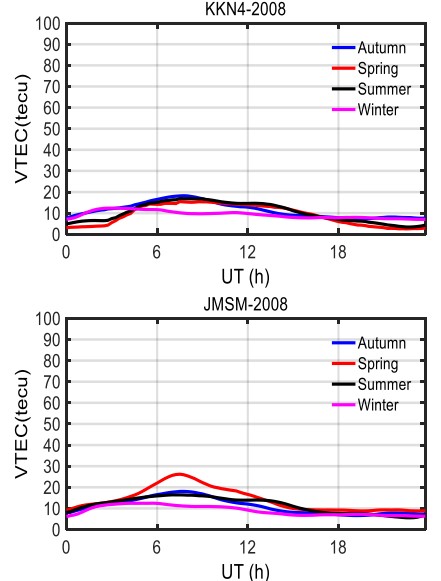
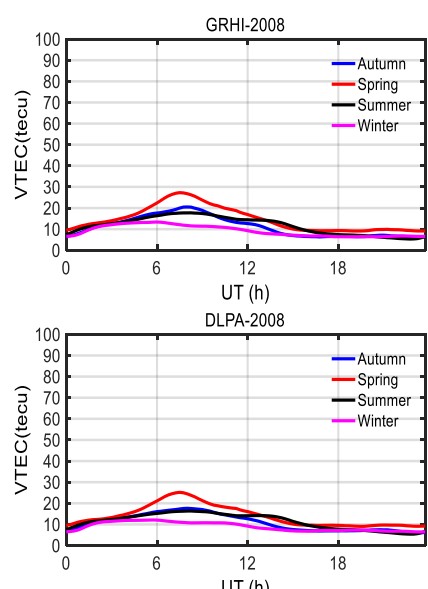





2    Figure 6b

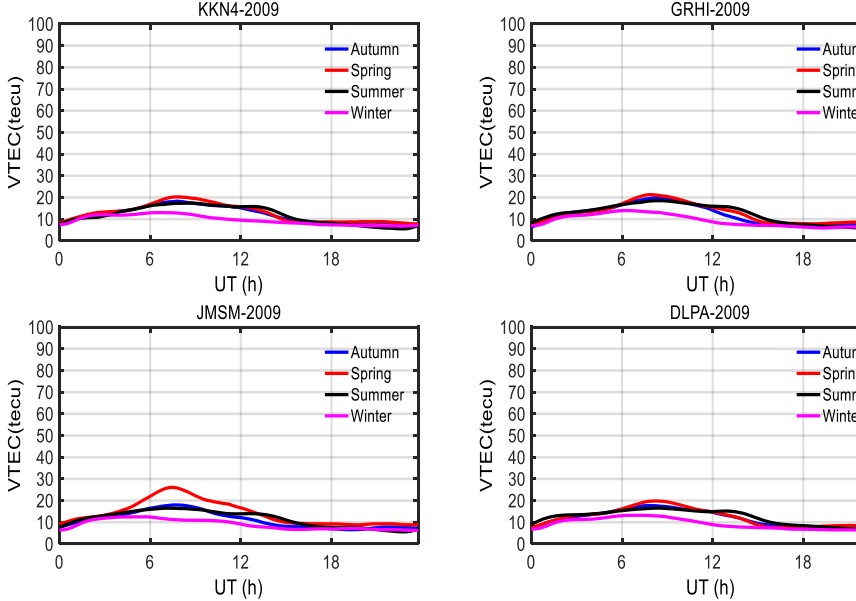





2    Figure 6c

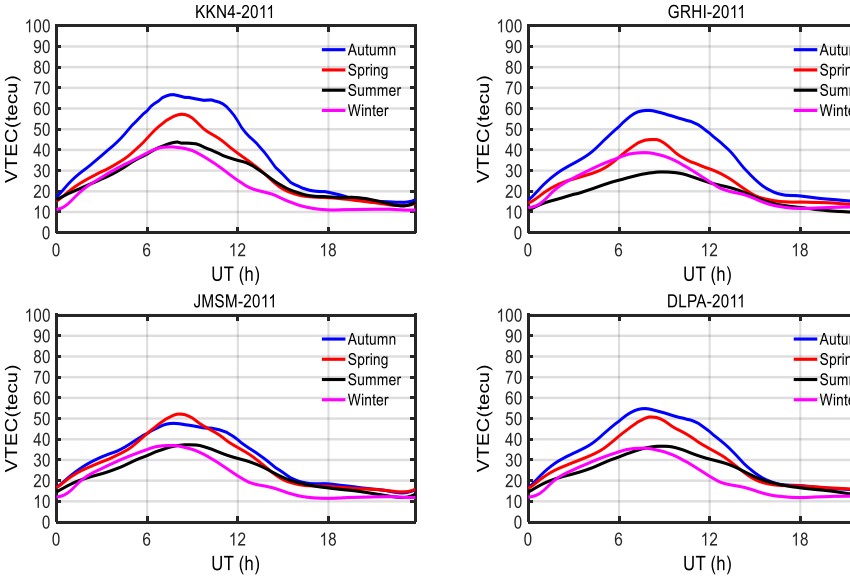





2     Figure 6d

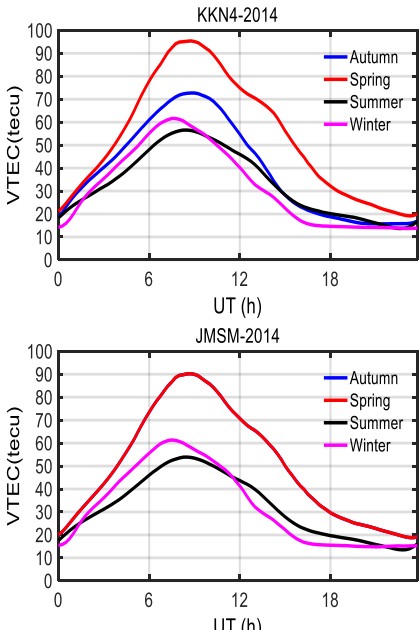

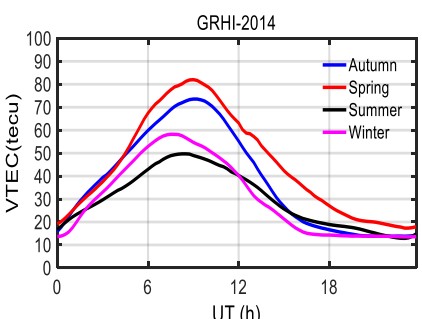





2    Figure 6e

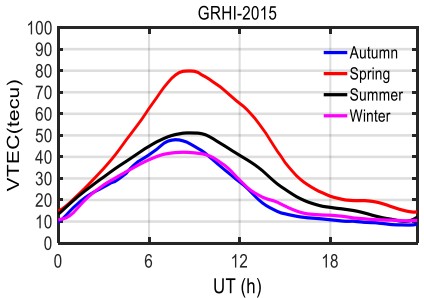

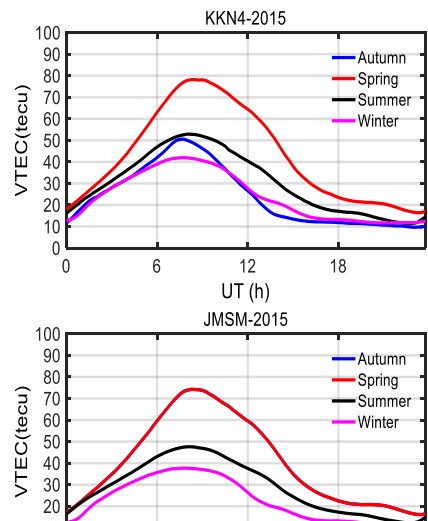





2    Figure 7

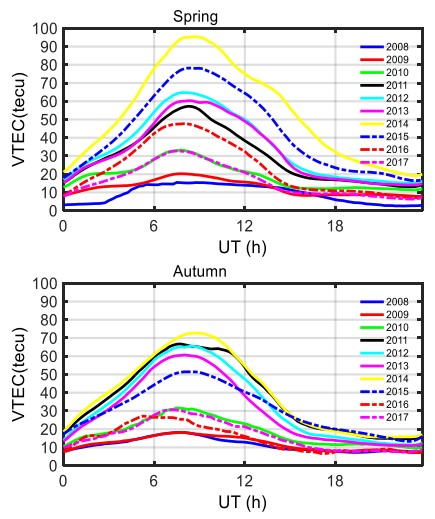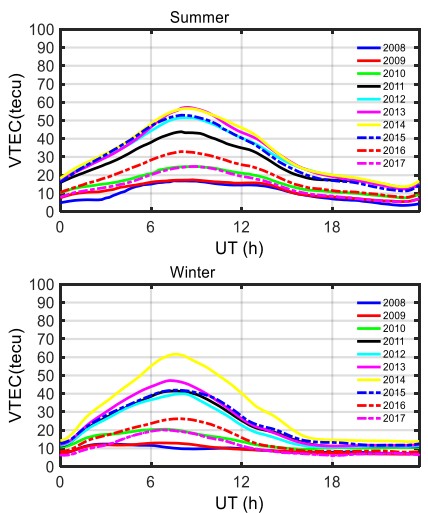





2     Figure 8

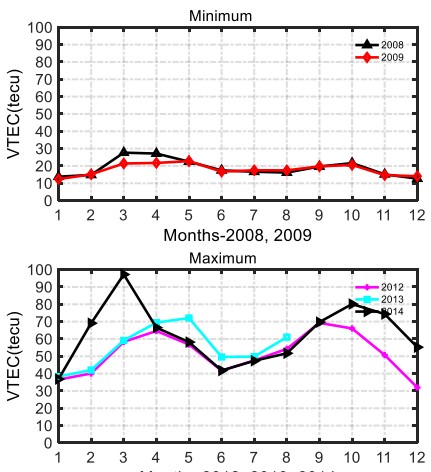

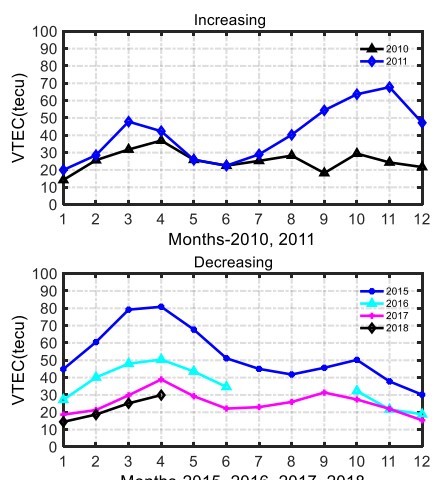



3    Figure 9

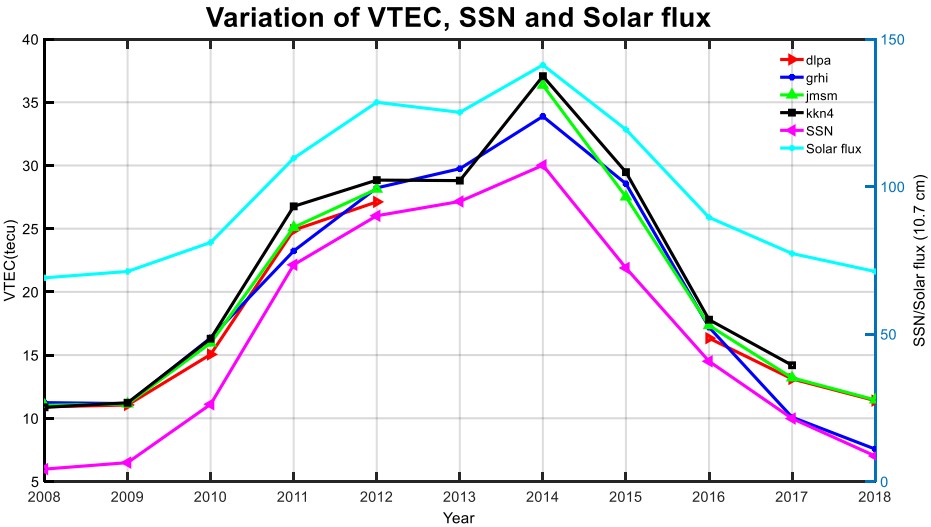

