# Peer review of "Climatology of ionosphere over Nepal based on GPS TEC data from 2008 to 2018"

_Annales Geophysicae, 2020_

## Referee Comment (RC1) · Anonymous Referee #1 · 12 Feb 2021

Summary The authors investigated the diurnal, monthly, annual, seasonal and solar cycle variations of VTEC at four locations in Nepal during the years 2008 to 2018. The results revealed a conventional diurnal variation pattern observed elsewhere and seasonal variation with maxima in spring equinox. Variability of TEC with solar activity was also reported. The authors claimed that this is the first report of a long-term study of the ionosphere over Nepal using TEC. This makes the work interesting and offers a good merit for publication.

Technical observations Page 2, Line 10: 'the' before 'most' should be deleted Page 2, Line 18-19: The sentence "Ouattara and Amory- Mazaudier (2012) showed solar activity occurring 18 different phases of solar cycle in diurnal variability" should be rewritten. Page 3, Line 9: Insert 'been' after 'have'

[Figure]

Page 6, line 28. Figure should always be written in full wherever is being referred to. It should also start with capital letter 'F'. Check throughout the text Page 6, line 25: the subsection should be '3.3'. then consider correcting other subsections 3.4, 3.5 etc Page 8, line 19: What is the significance of subsection 3:4- "Effects of geomagnetic activity on TEC"?. The authors may have to delete the entire section

General observations The result section on seasonal variation has some elements of discussion while other subsections have only presentation of results.

The author should have combined their results and discussions together in a single section for easy comprehension. The present format allows the discussion to be over-loaded with basic physics and thus make it a little complicated to understand.

The appropriate items under section 4 tagged 'Discussions' should be engaged to discuss the appropriate results itemized under section 3. Section 3 should be re-titled as ' Results and discussions'

Please also note the supplement to this comment:
https://angeo.copernicus.org/preprints/angeo-2020-82/angeo-2020-82-RC1-supplement.pdf

---

## Referee Comment (RC2) · Anonymous Referee #2 · 18 Feb 2021

Authors present a solar cycle-long climatological study of the VTEC performance above Nepal region. Authors present different scale variabilities – from the diurnal patterns through seasonal and annual to the solar cycle ones. Authors also make references to the physical mechanisms beyond the different patterns. I would recommend publication after adressing few minor/moderate issues:

Authors summarise the general objective of the work as climatology over Nepal, that has never done before, on the other hand they point out that similar studies for the Asian region has been done before. Authors address those studies in discussion, but it would be worth to precise what is the Nepalese climatology research area in this field during the manuscript objective formulation.

p. 5, l. 4 – Authors' statement about GIMs resolution (only 1 or 2 hours available) is

inaccurate – there are 15-minute GIMs available in the IGS repository (e.g. UQRG).

p. 6, l. 7-8 and figure 3b – Authors distinguish diurnal patterns of VTEC variability into parabolic and wave-like ones, however the separation of years 2016 and 2017 is questionable – in Figure 3b the profiles for 2016 and 2017 look pretty similar.

p. 6, l. 14 – Authors discuss minimum during solstices, but address only January solstice.

The results of monthly and seasonal variability analyzes seem to lead to convergent conclusions. It should be checked and properly addressed, if there are any differences revealed between monthly and seasonal patterns.

The very last conclusion of the manuscript about Nepal-specific behavior of certain seasonal variabilities seems a bit exaggerated, as the manuscript does not provide or address any exact results for the other regions for a clear Nepal-specification distinction.

In the equation 3 and its description the elvation angle symbol looks like logical set membership operator rather than Greek epsilon.

---

## Author Comment (AC1) · 4 Mar 2021

We thank the reviewer for his comments and take into account all the comments on

Climatology of ionosphere over Nepal based on GPS TEC data from 2 2008 to 2018

MS No.: angeo-2020-82

Summary The authors investigated the diurnal, monthly, annual, seasonal and solar cycle variations of VTEC at four locations in Nepal during the years 2008 to 2018. The results revealed a conventional diurnal variation pattern observed elsewhere and seasonal variation with maxima in spring equinox. Variability of TEC with solar activity was also reported. The authors claimed that this is the first report of a long-term study of the ionosphere over Nepal using TEC. This makes the work interesting and offers a

good merit for publication.

Technical observations Page 2, Line 10: 'the' before 'most' should be deleted Ans: It is corrected

Page 2, Line 18-19: The sentence "Ouattara and Amory- Mazaudier (2012) showed solar activity occurring 18 different phases of solar cycle in diurnal variability" should be rewritten. Ans: It is corrected as Ouattara and Amory- Mazaudier (2012) showed impact of solar activity on diurnal variability during different phases of solar cycle.

Page 3, Line 9: Insert 'been' after 'have' Ans: It is corrected

Page 6, line 28. Figure should always be written in full wherever is being referred to. It should also start with capital letter 'F'. Check throughout the text Ans: It is corrected

Page 6, line 25: the subsection should be '3.3'. then consider correcting other subsections 3.4, 3.5 etc Ans: As per suggestions of referees the sections of result and discussions which was written separately are written under a single section as "Results and Discussion".

Page 8, line 19: What is the significance of subsection 3:4- "Effects of geomagnetic activity on TEC"?. The authors may have to delete the entire section Ans: It is deleted in the text.

General observations The result section on seasonal variation has some elements of discussion while other subsections have only presentation of results. Ans: Each section of results is made uniform by placing discussion followed by result.

The author should have combined their results and discussions together in a single section for easy comprehension. The present format allows the discussion to be overloaded with basic physics and thus make it a little complicated to understand. Ans: The results and discussion are managed under the same section to make the finding clear.

The appropriate items under section 4 tagged 'Discussions' should be engaged to
discuss the appropriate results itemized under section 3. Section 3 should be re-titled as ' Results and discussions'. Ans: The discussion is tagged with result in the text.

Please also note the supplement to this comment:
https://angeo.copernicus.org/preprints/angeo-2020-82/angeo-2020-82-AC1-supplement.pdf

―――――――――――――――――――――

---

## Author Comment (AC2) · 4 Mar 2021

We thank the reviewer for his comments and take into account all the comments on

Climatology of ionosphere over Nepal based on GPS TEC data from 2 2008 to 2018

MS No.: angeo-2020-82

Authors present a solar cycle-long climatological study of the VTEC performance above Nepal region. Authors present different scale variabilities – from the diurnal patterns through seasonal and annual to the solar cycle ones. Authors also make references to the physical mechanisms beyond the different patterns. I would recommend publication after adressing few minor/moderate issues:

[Figure]

Authors summarise the general objective of the work as climatology over Nepal, that has never done before, on the other hand they point out that similar studies for the Asian region has been done before. Authors address those studies in discussion, but it would be worth to precise what is the Nepalese climatology research area in this field during the manuscript objective formulation. Ans: The precise area of research is to study very first time the Ionosphere above Nepal using data of a solar cycle 24.

p. 5, l. 4 – Authors' statement about GIMs resolution (only 1 or 2 hours available) is inaccurate – there are 15-minute GIMs available in the IGS repository (e.g. UQRG). Ans: In 1998 and under IGS, the first maps of the TEC were with a few analysis centers and at 2 hour intervals (odd hours) due to a limited number of measurement in the network. The best global coverage has made it possible to increase the temporal rate of the maps with the calculation on even hours ensuring daily continuity at midnight. The current situation is as follows with the 8 independent analysis centers (on CDDIS website) Analysis center Number of maps per day Time interval between maps CASG 49 30 min CODG 25 1h ESAG 25 1h EMRG 25 1h JPLG 13 2h UPCG 13 2h UQRG 97 15 min WHUG 13 2h IGSG 13 2h

The interval varying between centers depends on the analysis software. The tendency is to provide 25 maps at 1 hour intervals even if the IGSG synthesis remains at 2 hours. Some centers offer a lower time period. Indeed, UQRG offers a prototype service at 15 minute intervals. In this article, our goal was not to study the different solutions offered. We have chosen to retain the experienced CODG analysis center, which has the longest annual series, and to take the IGSG maps which provide a summary of some different TEC results.

p. 6, l. 7-8 and figure 3b – Authors distinguish diurnal patterns of VTEC variability into parabolic and wave-like ones, however the separation of years 2016 and 2017 is questionable – in Figure 3b the profiles for 2016 and 2017 look pretty similar. Ans: The superimposed plot of year 2016 and 2017 show parabolic type of variation in VTEC during both years (Figure is attached below). It is corrected in the text.

p. 6, l. 14 – Authors discuss minimum during solstices, but address only January solstice. Ans: It is observed from the plot that TEC is minimum in both solstices (January and June)

The results of monthly and seasonal variability analyzes seem to lead to convergent conclusions. It should be checked and properly addressed, if there are any differences revealed between monthly and seasonal patterns. Ans: The difference between monthly and seasonal variation is not observed.

The very last conclusion of the manuscript about Nepal-specific behavior of certain seasonal variabilities seems a bit exaggerated, as the manuscript does not provide or address any exact results for the other regions for a clear Nepal-specification distinction. Ans: This line written below is omitted from the text. It is probably a characteristic of Nepal

In the equation 3 and its description the elvation angle symbol looks like logical set membership operator rather than Greek epsilon. Ans: It is corrected

Please also note the supplement to this comment:
https://angeo.copernicus.org/preprints/angeo-2020-82/angeo-2020-82-AC2-supplement.pdf

———————————————————

KKN4

— 2016
— 2017

**Fig. 1.** showing similarity between plot of 2016 and 2017

[Figure]

| Analysis center | Number of maps per day | Time interval between maps |
|---|---|---|
| CASG | 49 | 30 min |
| CODG | 25 | 1h |
| ESAG | 25 | 1h |
| EMRG | 25 | 1h |
| JPLG | 13 | 2h |
| UPCG | 13 | 2h |
| UQRG | 97 | 15 min |
| WHUG | 13 | 2h |
| IGSG | 13 | 2h |

**Fig. 2.** showing different analysis centre has different time interval

---

## Author Response (AR1)

We thank the reviewer for his comments and take into account all the comments

Authors present a solar cycle-long climatological study of the VTEC performance above Nepal region. Authors present different scale variabilities – from the diurnal patterns through seasonal and annual to the solar cycle ones. Authors also make references to the physical mechanisms beyond the different patterns. I would recommend publication after addressing few minor/moderate issues:

Authors summarize the general objective of the work as climatology over Nepal, that has never done before, on the other hand they point out that similar studies for the Asian region has been done before.
Authors address those studies in discussion, but it would be worth to precise what is the Nepalese climatology research area in this field during the manuscript objective formulation. The precise area of research is to study very first time the Ionosphere above Nepal using data of a solar cycle 24.

p. 5, l. 4 – Authors' statement about GIMs resolution (only 1 or 2 hours available) is inaccurate – there are 15-minute GIMs available in the IGS repository (e.g. UQRG).

In 1998 and under IGS, the first maps of the TEC were with a few analysis centers and at 2 hour intervals (odd hours) due to a limited number of measurement in the network. The best global coverage has made it possible to increase the temporal rate of the maps with the calculation on even hours ensuring daily continuity at midnight. The current situation is as follows with the 8 independent analysis centers (on CDDIS website)

| Analysis center | Number of maps per day | Time interval between maps |
|---|---|---|
| CASG | 49 | 30 min |
| CODG | 25 | 1h |
| ESAG | 25 | 1h |
| EMRG | 25 | 1h |
| JPLG | 13 | 2h |
| UPCG | 13 | 2h |
| UQRG | 97 | 15 min |
| WHUG | 13 | 2h |
| IGSG | 13 | 2h |

The interval varying between centers depends on the analysis software. The tendency is to provide 25 maps at 1 hour intervals even if the IGSG synthesis remains at 2 hours. Some centers offer a lower time period. Indeed, UQRG offers a prototype service at 15 minute intervals. In this article, our goal was not to study the different solutions offered. We have chosen to retain the experienced CODG analysis center, which has the longest annual series, and to take the IGSG maps which provide a summary of some different TEC results.

p. 6, l. 7-8 and figure 3b – Authors distinguish diurnal patterns of VTEC variability into parabolic and wave-like ones, however the separation of years 2016 and 2017 is questionable – in Figure 3b the profiles for 2016 and 2017 look pretty similar.
The superimposed plot of year 2016 and 2017 show parabolic type of variation in VTEC during both years.
It is corrected in the text.

[Figure]

We added a comment on figure 3a
Mountains generate relief waves which propagate to stratosphere and lower thermosphere (Martin Leutbecher and Hans Volkert, 2000). Studies on these waves have been made in Nepal in the lower atmosphere (Regmi, R.P. and S. Maharjan, 2015; Regmi et al., 2017). Other studies have shown the impact of relief waves on the ionosphere in the Andes (Torre et al., 2014) and Tibet (Khan A., S. Jin, 2018). In the figure 3a we see oscillations which cannot be interpreted directly as the signature of the waves. In fact, for the processing of GPS data, we use pseudo-range signals which can be affected by reflections on surrounding reliefs as well as by waves.
With 5 references

Khan A., S. Jin, Gravity wave activities in Tibet observed by Cosmic GPS radio, Geodesy and Geodynamics 9 (2018) 504-511, https://doi.org/10.1016/j.geog.2018.09.009
Leutbecher Martin and Hans Volkert (2000), The Propagation of Mountain Waves into the Stratosphere: Quantitative Evaluation of Three-Dimensional Simulations, Journal of the Atmospheric Sciences Volume 57(18), DOI: https://doi.org/10.1175/1520-0469(2000)057<3090:TPOMWI>2.0.CO;2, 3090–3108
Torre, P. Alexander, P. Llamedo, R. Hierro, B. Nava, S. Radicella, T. Schmidt, and J. Wickert (2014), Wave activity at ionospheric heights above the Andes Mountains detected from FORMOSAT3/COSMIC GPS radio occultation data, Journal of Geophysical Research: Space Physics, 119(3):2046-2051
Regmi, R.P. and S. Maharjan, Trapped mountain wave excitations over the Kathmandu valley, Nepal, Asia-Pacific Journal of Atmospheric Sciences volume 51, pages303–309(2015)

Regmi, R.P., T. Kitada, J. Dudha, S. Maharjan, Large-scale gravity over the middle hills of the Nepal Himalayas: implication for aircraft accidents, Journal of applied meteorology and climatology, pp 371-389, February 2017, DOI: 10.1175/JAMC-D-160073.1

p. 6, l. 14 – Authors discuss minimum during solstices, but address only January solstice.
It is observed from the plot that TEC is minimum in both solstices (January and June)

The results of monthly and seasonal variability analyzes seem to lead to convergent conclusions. It should be checked and properly addressed, if there are any differences revealed between monthly and seasonal patterns.
The difference between monthly and seasonal variation is not observed.

The very last conclusion of the manuscript about Nepal-specific behavior of certain seasonal variabilities seems a bit exaggerated, as the manuscript does not provide or address any exact results for the other regions for a clear Nepal-specification distinction.
This line is omitted from the text.

We added:
The highest Himalayan mountains on earth in Nepal, are the source of landform waves that travel through the stratosphere and the lower thermosphere where they deposit their energy and give birth to secondary gravity waves that can affect VTEC. In our climatology study we analyze average behaviors that do not allow the study of these waves. Another study analyzing individually each day and using phase processing of GPS signals should be done in the future to analyze the impact of the Himalayas on VTEC  and the impact of the low atmosphere on VTEC.

In the equation 3 and its description the elevation angle symbol looks like logical set membership operator rather than Greek epsilon.

It is corrected

---

## Referee Report (RR1)

**Comment on manuscript entitled "Climatology of ionosphere over Nepal based on GPS TEC data from 2008 to 2018" by Pandit et al.**

**This work is publishable subject to modifications according to comments given in following.**

Comment 1; Reviewer is convinced with replies against previous comments by another reviewer

Some additional comments need to be answered:

**Abstract:**

Line 27-28: "The maximum value of TEC is observed during spring 28 equinoxes than autumn equinox with a few anomalies" Replace anomalies by asymmetries.

Line 28-30: It is now well established that winter anomaly is only phenomena of F2 peak. It is not a feature of seasonal variation in TEC. So modify sentence like-seasonal variation in TEC is observed to be a manifestation of variation of solar flux, particularly the level of solar flux in consecutive solstices. Also discuss paper https://doi.org/10.1007/s10291-018-0795-x in result section to explain seasonal variation in TEC. This paper clarifies misconception of winter anomaly in TEC.

**Introduction Section:**

Add following references from the Indian longitude sector

1. https://doi.org/10.1007/s10509-019-3701-2
2. https://doi.org/10.1002/2014JA020559

**Methodology Section**

Reframe title from "Data sets and data analysis" to Dataset

Everything else seems good.

**Results and Discussion**

Line 15-19 should be

"In this section, we present the diurnal, monthly, seasonal, solar cycle and geomagnetic variation in GPS TEC over Nepal during the solar cycle-24. Figure 1 represents the position of chosen GPS stations in Nepal for this study and Figure 2 represents the variation of sunspot number and solar flux during the period 2008-2018."

Line 23: Increasing should be inclining and decreasing should be declining.

Line 4-5, page 6: remove following sentences or modify accordingly

"Diurnal variation of VTEC was studied by plotting similar 5 curves for all the days from year 2008 to 2018 for all chosen four stations."

Line 19 page 7; Replace tecu by TECU, do similar changes throughout the paper.

Line 5-6 Page 8; Nepal sometimes shows features of EIA crest latitude and sometimes EIA trough stations.

This is totally wrong. Nepal is off crest region and lie in low-mid latitude region. It is not possible to exist in equatorial trough region. Remove sentence. Author should aware that magnetic equator is shifting towards south year by year.

Page 8: Reframe the text about winter anomaly following the comment given for abstract section.

**Conclusion section**

**Line 25-26;** "no diurnal 26 peak is observed during minimum and descending phases of the solar cycle"

This is not true, modify it.

Remove word "winter anomaly" from text following comment given against abstract section.

Revise whole conclusion section and write it in very precise and systematic manner.

Good Luck!

---

## Author Response (AR3)

**We thank the reviewer for his comments and take into account all the comments on**

**Comment on manuscript entitled "Climatology of ionosphere over Nepal based on GPS TEC data from 2008 to 2018" by Pandit et al.**

**This work is publishable subject to modifications according to comments given in following.** Comment 1; Reviewer is convinced with replies against previous comments by another reviewer Some additional comments need to be answered:

**Abstract:**

Line 27-28: "The maximum value of TEC is observed during spring 28 equinoxes than autumn equinox with a few anomalies" Replace anomalies by asymmetries.
It is replaced in the text, see line 28 page 1

Line 28-30: It is now well established that winter anomaly is only phenomena of F2 peak. It is not a feature of seasonal variation in TEC. So modify sentence like-seasonal variation in TEC is observed to be a manifestation of variation of solar flux, particularly the level of solar flux in consecutive solstices. Also discuss paper https://doi.org/10.1007/s10291-018-0795-x in result section to explain seasonal variation in TEC. This paper clarifies misconception of winter anomaly in TEC.
The sentence is replaced and this referred paper
Rao, S.S., Sharma, S., Pandey, R. (2019b),  Study of solar flux dependency of the winter
        anomaly in GPS TEC, *GPS Solutions*, **23**(4) DOI: 10.1007/s10291-018-0795-x
 is used to explain seasonal variation in the text see lines 14-18 page 9

**Introduction Section:**

Add following references from the Indian longitude

    sector 1. https://doi.org/10.1007/s10509-019-3701-

Rao, S.S., Chakraborty, M., Kumar, S., Singh, A.K. (2019a). Low-latitude ionospheric response
        from GPS, IRI and TIE-GCM TEC to Solar Cycle 24, *Astrophys Space Sci*, **364**:216,
        2. https://doi.org/10.1002/2014JA020559
Dashora, N., and Suresh, S. (2015). Characteristics of low-latitude TEC during solar cycles 23
        and 24 using global ionospheric maps (GIMs) over Indian sector, *J. Geophys. Res. Space
        Physics,* **120**, 5176–5193, doi:10.1002/2014JA020559. A

    These  references are added in the text, see lines 6-10 page 3

**Methodology Section**

Reframe title from "Data sets and data analysis" to

Dataset Everything else seems good.

It is changed in the text, see line 27 page 3

**Results and Discussion**

Line 15-19 should be

"In this section, we present the diurnal, monthly, seasonal, solar cycle and geomagnetic variation in GPS TEC over Nepal during the solar cycle-24. Figure 1 represents the position of chosen GPS stations in Nepal for this study and Figure 2 represents the variation of sunspot number and solar flux during the period 2008-2018."
It is replaced in the text, see lines 21- 24 page 5

Line 23: Increasing should be inclining and decreasing should be declining.
It is changed in the text, see line 2 page 6

Line 4-5, page 6: remove following sentences or modify accordingly

"Diurnal variation of VTEC was studied by plotting similar 5 curves for all the days from year 2008 to 2018 for all chosen four stations."
It is modified in the text, see lines 14-16 page 6

Line 19 page 7; Replace tecu by TECU, do similar changes throughout the paper.

It is replaced in the all the text

Line 5-6 Page 8; Nepal sometimes shows features of EIA crest latitude and sometimes EIA trough stations.
This sentence is omitted from the text; see lines 16-18 page 8

This is totally wrong. Nepal is off crest region and lie in low-mid latitude region. It is not possible to exist in equatorial trough region. Remove sentence. Author should aware that magnetic equator is shifting towards south year by year.

Page 8: Reframe the text about winter anomaly following the comment given for abstract section.
The text is reframed; see lines 3-6 and 11-14 page 9

**Conclusion section**

**Line 25-26;** "no diurnal 26 peak is observed during minimum and descending phases of the solar cycle"

This is not true, modify it.

It is corrected in the text; see line 15 page 13

Remove word "winter anomaly" from text following comment given against abstract section.

The word "winter anomaly" has removed from the text.

Revise whole conclusion section and write it in very precise and systematic manner.

The conclusions are arranged systematically and precisely; see lines 27-30 page 13 and lines 1-2 page 14.

Good Luck!

[revised manuscript text omitted]

In this study, we use GPS derived TEC from RINEX file using this method to obtain TEC calibrated at 15 minutes for all measures. Between 30s VTEC sequences, the elevation may vary. This leads to variation in the VTEC depending on the constellation and not just the variation of the content over that period. We have chosen to do the regression over a period 15 minutes with the VTEC obtained displayed in the middle of this period. This makes it possible to have 4 points over 1 hour and therefore to have an evolution of the VTEC 4 times more precise than that of GIM maps which are currently in steps 1 or 2 hours depending on the organization. So, it is better possibility to see and characterize finer local structures in RINEX derived TEC than in GIM.

This study analyzes variations of VTEC during different phases of solar cycle 24 along with annual, seasonal and diurnal variation. For this local season classified as winter (November, December, January and February), spring (March and April), summer (May, June, July and August) and autumn (September and October). The classifications of selected years as per solar cycle phases are presented in table 2.

**3. Results and Discussion**

In this section, we present the diurnal, monthly, seasonal, solar cycle and geomagnetic variation in GPS TEC over Nepal during the solar cycle-24. Figure 1 represents the position of chosen GPS stations in Nepal 
[revised manuscript text omitted]
 2011  the VTEC is larger in Winter than in Summer at KKN4 whereas GRHI and JMSM the Winter VTEC is  smaller than the Summer one in year 2011 and 2014.  At DLPA, the Winter VTEC is not larger than the Summer VTEC. In year 2008 spring VTEC identified more than autumn for GRHI, JMSM and DLPA but less than autumn is observed KKN4. In year 2009 only at JMSM spring noticed greater than autumn. The autumn VTEC is greater than spring for all station in 2011 except at JMSM it equal to spring. Large asymmetry is noticed between spring and autumn in year 2014. In year 2015 the summer peak is higher than autumn. In present study the VTEC larger in winter than VTEC in Summer  is noticed in 2011 and 2014. At KKN4 station the VTEC larger in winter than VTEC in Summer  is noticed in the year 2014; at GRHI 2014 and 2016; at JMSM 2014 and 2016. The VTEC larger in winter than Summer VTEC  is not noticed at DLPA (Figure 6c and 6d).

Solar flux dependency of winter anomaly in GPS TEC has studied by Rao et al. (2019b). The result showed that when the level of solar flux in winter month is greater than the corresponding summer month winter anomaly is observed irrespective to the phases of solar cycle whether it is high or low. Their study also pointed out that the winter anomaly in GPS-derived TEC may not be a feature of any geophysical significance. 
[revised manuscript text omitted]

Dashora, N., and Suresh, S. (2015). Characteristics of low-latitude TEC during solar cycles 23 and 24 using global ionospheric maps (GIMs) over Indian sector, *J. Geophys. Res. Space Physics,* **120**, 5176–5193, doi:10.1002/2014JA020559.

deAbreu, A. J., Fagundes, P.R., Gende, M., Bolaji, O. S., de Jesus, R., Brunini, C. (2014). Investigation of ionospheric response to two moderate geomagnetic storms using GPS-TEC measurements in the South American and African sectors during the ascending phase of solar cycle 24, *Advances in Space Research*, **53** (9), 1313–1328, doi:10.1016/j.asr.2014.02.011.

[revised manuscript text omitted]

Rao, S.S., Chakraborty, M., Kumar, S., Singh, A.K. (2019a). Low-latitude ionospheric response
from GPS, IRI and TIE-GCM TEC to Solar Cycle 24, *Astrophys Space Sci*, **364**:216,
https://doi.org/10.1007/s10509-019-3701-2

Rao, S.S., Sharma, S., Pandey, R. (2019b).Study of solar flux dependency of the winter anomaly
in GPS TEC, *GPS Solutions*, **23**(4) DOI: 10.1007/s10291-018-0795-x

[revised manuscript text omitted]

Figure 2

[Figure]

Figure 3a

[Figure]

Figure 3b

[Figure]

[Figure]

Figure 4

[Figure]

Figure 5a

[Figure]

Figure 5b

[Figure]

Figure 5c

[Figure]

Figure 5d

[Figure]

Figure 6a

[Figure]

Figure 6b

[Figure]

Figure 6c

[Figure]

Figure 6d

[Figure]

[Figure]

Figure 6e

[Figure]

Figure 7

[Figure]

Figure 8

[Figure]

Figure 9

[Figure]